# E-Nose Discrimination of Abnormal Fermentations in Spanish-Style Green Olives

**DOI:** 10.3390/molecules26175353

**Published:** 2021-09-02

**Authors:** Ramiro Sánchez, Elísabet Martín-Tornero, Jesús Lozano, Emanuele Boselli, Patricia Arroyo, Félix Meléndez, Daniel Martín-Vertedor

**Affiliations:** 1Technological Institute of Food and Agriculture (CICYTEX-INTAEX), Junta of Extremadura, Avda. Adolfo Suárez s/n, 06007 Badajoz, Spain; ramiro.sanchez@juntaex.es; 2Department of Agricultural and Forestry Engineering, School of Agrarian Engineering, University of Extremadura, 06007 Badajoz, Spain; elisabetmt@unex.es; 3Industrial Engineering School, University of Extremadura, 06006 Badajoz, Spain; jesuslozano@unex.es (J.L.); parroyoz@unex.es (P.A.); felixmv@unex.es (F.M.); 4Research Institute of Agricultural Resources (INURA), Avda de la Investigación s/n, Campus Universitario, 06006 Badajoz, Spain; 5Faculty of Science and Technology, Free University of Bozen-Bolzano, Piazza Università 1, 39100 Bolzano, Italy

**Keywords:** sensory analysis, volatile compounds, defects, e-nose, table olives

## Abstract

Current legislation in Spain indicates that table olives must be free of off-odors and off-flavors and without symptoms of ongoing alteration or abnormal fermentations. In this regard, the International Olive Council (IOC) has developed a protocol for the sensory classification of table olives according to the intensity of the predominantly perceived defect (PPD). An electronic nose (e-nose) was used to assess the abnormal fermentation defects of Spanish-style table olives that were previously classified by a tasting panel according to the IOC protocol, namely zapateria, butyric, putrid, and musty or humidity. When olives with different defects were mixed, the putrid defect had the greatest sensory impact on the others, while the butyric defect had the least sensory dominance. A total of 49 volatile compounds were identified by gas chromatography, and each defect was characterized by a specific profile. The e-nose data were analyzed using principal component analysis (PCA) and partial least square discriminant analysis (PLS-DA). The different defects were clearly separated from each other and from the control treatment, independently of PPD intensity. Moreover, the e-nose differentiated control olives from table olives with combined sensory defects despite the dilution effect resulting from the combination. These results demonstrate that e-nose can be used as an olfactory sensor for the organoleptic classification of table olives and can successfully support the tasting panel.

## 1. Introduction

Spain is the largest producer of table olives in the world, representing a market that generates annual trade valued at 1.7 billion euros worldwide. Olives produced in Spain are present in almost all countries. Moreover, the export of table olives to other countries, such as Morocco, Egypt, Greece, Turkey, Argentina, Peru, and Portugal, has increased in recent years [1].

To produce table olives, the “Spanish style” is the most common process. The fruits of *Olea europaea* spp. are treated with caustic soda to remove their bitterness and then fermented in a salt solution for several months. However, a number of critical points during the elaboration process can facilitate abnormal fermentation, thereby leading to defective olives. These alterations are among the main causes of economic loss for producers.

The current legislation only considers the physical defects of extra or fancy, first choice or selected olives, and second choice or standard olives [2]. However, the regulation indicates that table olives must be free of strange odors and flavors and free of symptoms of ongoing alteration or abnormal fermentation. Therefore, to classify table olive defects according to sensory analysis, producers should analyze samples with a tasting panel trained and validated by the IOC (2011) [3]. However, the protocol established by the IOC is only a recommendation and is not yet in force. The sensory panel should classify table olives according to olfactory defects, including putrid, zapateria, butyric, musty, rancid, or vinegary sensations [4,5]. These defects perceived by tasters are normally present in table olives, in particular the zapateria defect. According to the IOC regulation, the zapateria defect is caused by a combination of volatile fatty acids formed by abnormal fermentation, leading to the sensation of rotten leather. The butyric defect is the off-flavor of rancid butter or cheese. The putrid defect is the odor of decaying organic matter. Finally, the musty or humidity defect produces a smell of mold. These defects are probably caused by bad industrial practices that facilitate uncontrolled development of the fermentation process.

Analysis of volatile compounds by chromatography may also help to identify compounds responsible for abnormal fermentations [6,7,8,9]. However, sensory analyses based on a trained expert panel and characterization of the volatile fraction of fermented olives based on gas chromatography are expensive, laborious, and time-consuming procedures requiring sophisticated equipment and/or skilled personnel. Thus, it is important to develop a fast and reliable technique to discriminate table olives according to their sensory characteristics. The same protocol should also be used to identify incipient defects in olives to control the onset of anomalous fermentations.

The electronic nose (e-nose) is an electromechanical powerful sensory device that enables the discrimination of aroma profiles of different matrices, such as wine [10], edible mushrooms [11], cooked chicken [12], edible oils [13], or fresh vegetables [14]. This device is also used to classify olives on olive trees [9] or even to differentiate table olives elaborated according to the Spanish-style protocol [7]. In this respect, the e-nose may be a fast, cheap, and effective alternative to identify different types of fermentation defects in large amounts of olives on an industrial scale. The e-nose is a nondestructive device complementary to the tasting panel. It can be routinely installed to identify early signs of off-flavors during the fermentation of Spanish-style table olives with the aim of correcting them before the olives become unacceptable and unmarketable.

The aim of the present study was to develop an analytical protocol based on an e-nose device to differentiate Spanish-style defective olives according to their sensory attributes. The data were compared with the profile of volatile compounds determined by gas chromatography and with sensory characterization of the olives carried out by a trained tasting panel.

## 2. Results and Discussion

### 2.1. Sensory Profile of Table Olives

The selected Spanish-style table olives were sensorially evaluated by a tasting panel in order to classify them according to the predominantly perceived defect (PPD). It should be noted that the quality standards [2] do not include sensory analysis as an evaluation criterion for classifying table olives into different commercial categories. This regulation only takes into account the physical defects in the fruit, such as softness, skin defects, or broken fruits. Therefore, in this research, the sensory evaluation and classification of the olives were obtained according to the IOC regulation based on the evaluation of PPD intensity by the tasting panel.

Healthy table olives were selected for the control treatment (control). However, several other olives presented various sensory defects related to abnormal fermentation. The main defects found in different samples were zapateria (D1), butyric (D2), putrid (D3), and musty or humidity (D4). Similarly, Marx et al. [15] evaluated several table olive samples using a tasting panel who obtained a similar classification of the samples according to the sensory defect detected (i.e., butyric, putrid, zapateria, musty, and/or winey–vinegary).

The tasters indicated that defects in the olives were of high intensity. For this reason, the olives were classified in the second or standard category because the PPD was higher than 3.5 and less than or equal to 6.0 (Table 1). Thus, all these olives could be legally marketed despite the significant defects [3].

Failure to control the product during the fermentation of Spanish-style olives causes an increase in pH, which can contribute to the development of microorganisms that cause abnormal fermentation due to their ability to metabolize lactic acid.

Combinations of the different olive defects were made to verify what the PPD was in the olives and to determine its intensity through the panel (Table 1). Thus, equally combined mixtures of defective olives were made as follows: zapateria + butyric (D1 + D2), butyric + putrid (D2 + D3), zapateria + putrid (D1 + D3), and zapateria + butyric + putrid (D1 + D2 + D3). As can be seen in Table 1, the intensity of the defect was reduced by almost half when the different combinations were made. Thus, a dilution effect was observed as the intensity of the defect decreased. When the defective olives were mixed, their commercial sensory category according to the IOC regulation improved as it went from the second category (3.5 < PPD ≤ 6.0) to the first category (2 < PPD ≤ 3.5). Mixing olives with different defects in the same package is a commercial strategy that allows companies to reduce the waste of defective olives. This practice is fully legal and allowed as long as the percentage of physical defects complies with the legislation.

When olives with different defects were mixed, one of the defects prevailed over the others (dominance effect). In fact, the putrid defect had the greatest sensory impact on the others, while the butyric defect had the least sensory dominance. Zapateria and musty defects presented an intermediate dominance. This result has interesting consequences for table olives producers. In the case of table olives with some defects developed in the fermentation tanks, an appropriate mixing of the olives according to the dominance of their defect can be useful to market olives of better quality that comply with the current legislation.

### 2.2. Volatile Compounds of the Pure Defects

Aroma is considered a quality index for olive products [16]. It is known that microorganisms play an important role in the formation of the volatile profile of fermented foods [16] and therefore play a decisive role in the characterization of the flavor profile of table olives.

The volatile compounds were analyzed in the five types of table olives (control, zapateria, putrid, butyric, and musty). The identified volatile compounds listed according to chemical group, odor attributes, and relative content in percentage of intact (control) and defective olives are shown in Table 2.

A total of 49 volatile compounds were identified. Among them, 20 compounds were found in healthy olives, 17 in zapateria and butyric samples, 20 in putrid samples, and 18 in musty samples.

Figure 1 shows the composition of the five types of samples for each chemical class.

The compounds isolated and identified in healthy control olives were mainly phenols (52%), while those identified in zapateria and butyric samples were mainly carboxylic acids (83% and 84%, respectively). On the other hand, putrid and musty samples showed a higher content of alcohols and other compounds, respectively, with respect to the healthy olives.

Regarding the individual compounds (Table 2), the major constituents of the volatile matrix in healthy samples were creosol (40.2%), a monoterpene derivative (15.3%), 2-ethyl-phenol (8.7%), phenylethyl alcohol (7.8%), benzoic acid (7.4%), and acetic acid (7.4%). Comparison with literature data on the volatile composition of olive fruits is difficult because of the great variability among different studies. Volatile compounds are, in fact, strongly influenced by many factors, such as variety, ripening state, or processing conditions [16,17,18]. However, these compounds decrease considerably in olives with defects, and other different compounds appear. In the present work, the main constituents of zapateria samples were butanoic acid (37.9%), (E)-3-hexenoic acid (17.8%), hexanoic acid (8.5%), and pentanoic acid (valeric acid) (5.4%). These results are in agreement with previous studies, which associated these short chain fatty acids with zapateria spoilage [8,19,20]. Cyclohexanecarboxylic acid was found only in samples with this defect. It did not represent a large proportion of the total identified volatile compounds (3.7%), but it has been identified as a key compound of zapateria samples in previous studies [19]. In other studies, it has been reported that cyclohexanecarboxylic acid, in combination with other volatile acids, appears to be responsible for the unpleasant smell typical of zapateria olives [21].

The major volatile compounds responsible for the butyric defect were butanoic acid (55.5%), pentanoic acid (18.5%), propanoic acid (4.9%), and butan-2-ol (7.6%). This result is in agreement with previous studies [20].

On the other hand, the putrid defect has a completely different volatile profile from the defects described above, as shown in Figure 1. In this case, the major volatile compounds were isopropyl alcohol (16.0%), phenylethyl alcohol (15.9%), propanoic acid (15.8%), 2,4-dimethyl-heptane (14.5%), and 3-methyl-butan-1-ol (9.3%). As far as we know, there are only very few studies describing the volatile composition of the putrid defect.

Finally, the defect indicated as musty had a volatile profile similar to the putrid defect. The major volatile compounds present in the olives affected by the musty defect were 2-methoxy-phenol (46.2%), followed by 2,4-dimethyl-heptane (23.6%) and styrene (8.7%). This is the first time that the volatile compounds of the musty defect were isolated and identified.

### 2.3. Discrimination of Table Olive Defects with the E-Nose

A multisensory system (e-nose) was used to classify table olives according to the sensory defect. The volatile compounds emanated from the samples were put in contact with the sensor array. The response of each sensor was a different instrumental signal with an amplitude that depended on its interaction with the sample. A radial graph (Figure 2) was drawn to show the different amplitude of the responses for the 11 sensors of the e-nose.

For the representation of the radial graph, the extracted features of each sensor were normalized using the criteria established by [7] to have all the data on the same magnitude scale. The radial profile for each sample was different depending on the type of alteration of the olives. The set of sensors gave different signals for the five groups of samples, suggesting that they all contributed to odor discrimination. The complexity of the output data necessitated the use of multivariate analysis methods, such as PCA, as discussed in the next sections.

### 2.4. Discrimination of Isolated Defects

The e-nose data of table olive samples with different defects were first analyzed by principal component analysis (PCA) (Figure 3) to obtain a better visualization of the interactions between the variables and grouping of the samples.

PCA is a well-known pattern-recognition technique, which returns results as a projection of the data into a reduced hyperspace defined by the principal components [22]. Principal components are linear combinations of the original variables, where the first principal component represents the largest variance, the second principal component accounts for the second largest variance, and so on.

The PCA results showed that 59.8% of the total variance of data was explained by PC1 and 28.2% by PC2. The model based on the first two components showed a clear differentiation of the samples according to their olfactory characteristics and was able to separate healthy olives from those with defect in the fermentation process.

The PCA of the data showed that the e-nose response well fitted with the sensory analysis performed by the trained panel.

After the good results obtained in the PCA, a classification analysis was performed using PLS-DA and leave-one-out cross-validation. The results are shown in Table 3 as a confusion matrix. The sum of the diagonal elements of the confusion matrix gives the percentages of correct predictions. As can be seen, about 99% of correct predictions was obtained.

These results show the ability and accuracy of e-nose to discriminate between different defects (zapateria, butyric, putrid, and musty) and compare them with the control treatment (healthy olives). Thus, the e-nose is able to discriminate olives according to their quality at an industrial level. This tool can be used to control the fermentation process of olives to ensure their quality.

### 2.5. Discrimination of Combined Defects

The response of the e-nose to odor patterns resulting from combinations of different fermentation defects was also studied. As in the previous case, the data were first analyzed using PCA. The score plot of the two first principal components for the discrimination of samples with combinations of defects is shown in Figure 4.

PCA based on e-nose data differentiated control olives from table olives with combined sensory defects despite the dilution effect resulting from the combination. The first and second principal components (PC1 and PC2) were sufficient to visualize the data structure as they explained 76% of the total variance.

Subsequently, PLS-DA was applied to construct a classification model and the corresponding confusion matrix (leave-one-out cross-validation), the results of which are shown in Table 4.

The results obtained (99% of correct predictions) showed that the combination of more than one defect and the dilution effect of the intensity of the odor pattern was not an obstacle for the e-nose; in fact, a clear discrimination was always evidenced. Furthermore, the e-nose also showed a clear differentiation between the control and combined defects, whose values of PPD are shown in Table 1 (control: PPD = n.d. (extra category) and defect samples that correspond to the first category (3 < PPD ≤ 4.5)). To the best of our knowledge, there is not much literature on the discrimination of defective olives with electronic devices. However, a study [23] described the prediction of the cooked defect produced by the application of different sterilization treatments in oxidized black olives from two olive varieties using an electronic tongue. Therefore, these results highlight the feasibility of these devices as rapid analytical tools to monitor the processing of table olives.

## 3. Materials and Methods

### 3.1. Table Olives Elaboration

Olives of the “Carrasqueña” variety were harvested at the green stage of ripeness within the limits of the “Tierra de Barros” olive-growing area (Badajoz, Spain) during the 2019/2020 campaign and were processed according to the Spanish-style protocol [24] by a company located in the southwest of Extremadura (Spain). The product was introduced into fermenters with the capacity of 236 L in three replications. During the fermentation process, aliquots of olives were sampled and a sensory analysis was carried out by a trained tasting panel. After completion of the fermentation process, the olives were covered with brine (4% *w/v* NaCl) and sealed in cans (150 g each can). Each week, the cans of olives were taken and a sensory analysis was performed by the same tasting panel with the aim of identifying Spanish-style table olives with abnormal fermentation defects (D1, D2, D3, and D4). When the olives showed the desired defect, they were kept in a refrigerator (4 °C) until the analysis was carried out. A control sample without fermentation defects was also stored (control).

The experiments were carried out in a standard glass jar containing as many olives as the bottom of the glass could hold and arranged in a single layer. Then, 10 mL of covering liquid was added on top of the olives following the IOC rules. In addition, to obtain the combinations of defects, samples were mixed proportionally. D1 + D2, D2 + D3, and D1 + D3 were mixed in a 50:50 ratio by mixing 2 olives and 5 mL of saline solution from each sample. For the combination of three defects, D1 + D2 + D3 were also mixed equally by taking 1 olive and 3 mL of each sample.

### 3.2. Analyses

The table olives were subjected to sensory analysis, characterization of volatile compounds, and e-nose measurements as detailed below.

#### 3.2.1. Sensory Analysis

Table olives were evaluated by a sensory panel composed of eight experts from the CICYTEX Research Center (Extremadura, Spain) who were trained according to the IOC recommendations [3]. For this study, the intensity and type of the off-odor perceived by the taster was assessed on a structured scale from 0 to 10. The results were expressed as median values of defects; values were considered valid when the coefficient of variation was less than 20. Finally, table olives were classified according to the quality categories established by the IOC [3].

One-way ANOVA was performed followed by Tukey’s multiple range test to establish statistically significant differences between the different samples. Significance was set at *p* < 0.05. SPSS 18.0 software was used for statistical analysis (SPSS Inc., Chicago, IL, USA). Data were expressed as mean and standard deviations (SD).

#### 3.2.2. Analysis of Volatile Compounds

The volatile compounds were analyzed in triplicate with a Bruker Scion 456-GC triple quadrupole gas chromatograph. Pitted olives were crushed and homogenized following the procedure reported in [25]. A 2 g aliquot was mixed with 7 mL of a 30% NaCl solution in a 15 mL glass vial. Volatile components were sampled from the headspace at 40 °C for 15 min using SPME with a polydimethylsiloxane/divinylbenzene (PDMS/DVB) StableFlex fiber (65 μm, Supelco). After SPME, desorption was carried out at the injection port of the gas chromatograph at 250 °C for 15 min. The components were separated using a VF-5MS capillary column (30 m × 0.25 mm; ID: 0.25 mm). The tentative identification of the analytes was based on comparison of mass fragmentations with the NIST 2.0 MS library.

#### 3.2.3. E-Nose Analysis

The e-nose equipment was a portable miniaturized device designed by the University of Extremadura (Spain) [26]. This prototype consisted of four digital gas sensor chips with integrated metal oxide (MOX) sensors: BME680 from Bosch, SGP30 from Sensirion, and CCS811 and iAQ-Core from ScioSense. The microprocessor read the values detected by the sensors, formatted them, and sent them to an external smart device via Bluetooth. The resulting data were then passed to a computer.

E-nose measurements were performed following IOC recommendations. Specifically, 10 mL of brine containing four olives was introduced into standard tasting glasses, covered with a watch glass, and placed on a block thermostatted at 25 °C. Another standard tasting glass was left without samples to serve as a baseline reference. Five measurements were taken for each table olive sample, and each data acquisition cycle consisted of two parts. First, the e-nose was placed on the sample glass for 60 s and the sensor signals were recorded. Then, the e-nose was moved to the glass without sample to perform desorption with free air for 30 s to bring the gas sensor signal back to the baseline.

Each sensor response curve consisted of N points corresponding to the sensor measurements with time. The features used to characterize the sensor response curves were the maximum signal value minus the minimum signal value multiplied by 100 and subtracted by 1 ((MAX-MIN)×100 − 1). As a result, a data vector with 11 rows (sensors) for each sample was obtained.

#### 3.2.4. Multivariate Data Analysis

The e-nose data consisted of a matrix of 100 rows (10 measurements for each duplicate sample) and 11 columns (sensors). The data were first subjected to principal component analysis (PCA) to perform an exploratory analysis. Subsequently, PLS-DA was applied to build the classification model. As the variables were measured in different units, the original variables were autoscaled. Data analysis was performed using Matlab R2016b version 9.1 (The Mathworks Inc., Natick, MA, USA) with PLS_Toolbox 8.2.1 (Eigenvector Research Inc., Wenatchee, WA, USA).

## 4. Conclusions

The e-nose proved to be a useful tool for recognizing olfactory sensations derived from abnormal fermentations occurring in table olives, such as zapateria, butyric, putrid, and musty defects. The classification made with the e-nose coincided with the results obtained by the tasting panel. When combined with chemometric tools (PCA and PLS-DA), e-nose provides a rapid and inexpensive method to monitor the occurrence of abnormal fermentations during the processing of table olives. Therefore, the e-nose can play a role in the table olive industry in the future for the rapid in-house classification of commercial table olives according to the criteria set by the IOC or when a tasting panel is not available. In addition, the panel leader can use the e-nose as a support to discriminate samples with different intensity of the predominant defect in order to produce table olives compositions of better quality that comply with current market legislation.

## Figures and Tables

**Figure 1 molecules-26-05353-f001:**
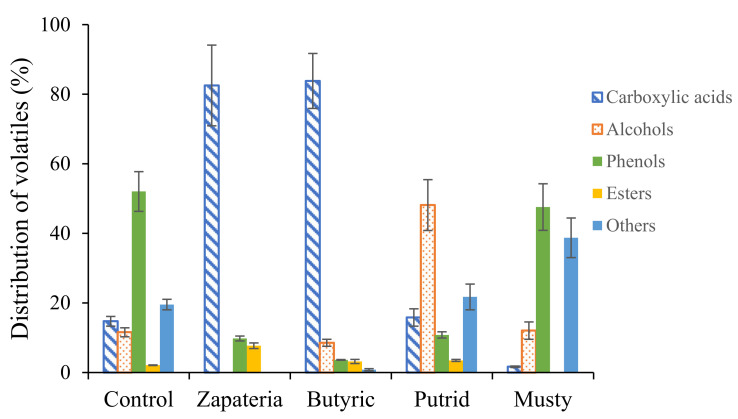
Distribution of chemical families of volatile compounds in healthy and defective olives.

**Figure 2 molecules-26-05353-f002:**
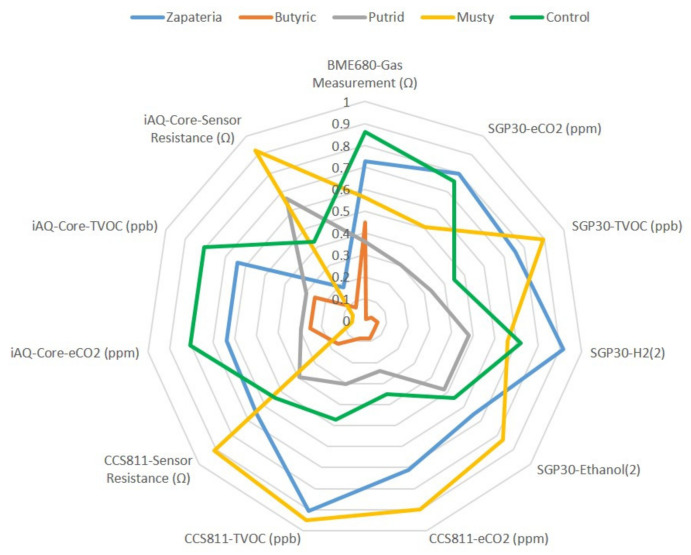
Radial plots for the responses of the sensor array to the control and defective table olives.

**Figure 3 molecules-26-05353-f003:**
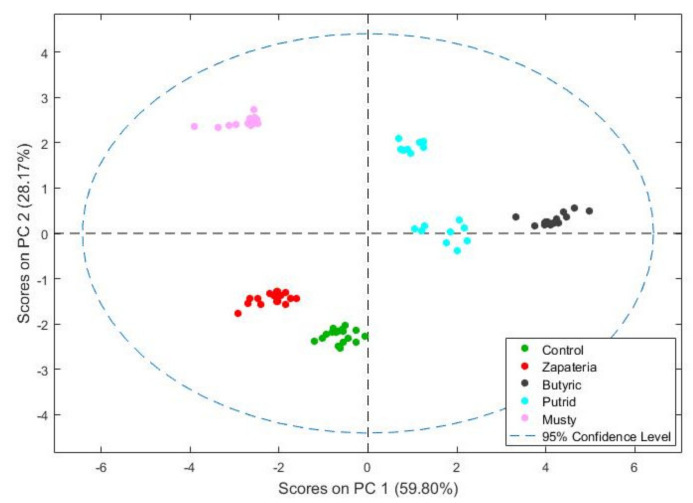
Score plot of the PCA analysis for healthy olives (control) and olives with off-odor of zapateria, butyric, putrid, and musty.

**Figure 4 molecules-26-05353-f004:**
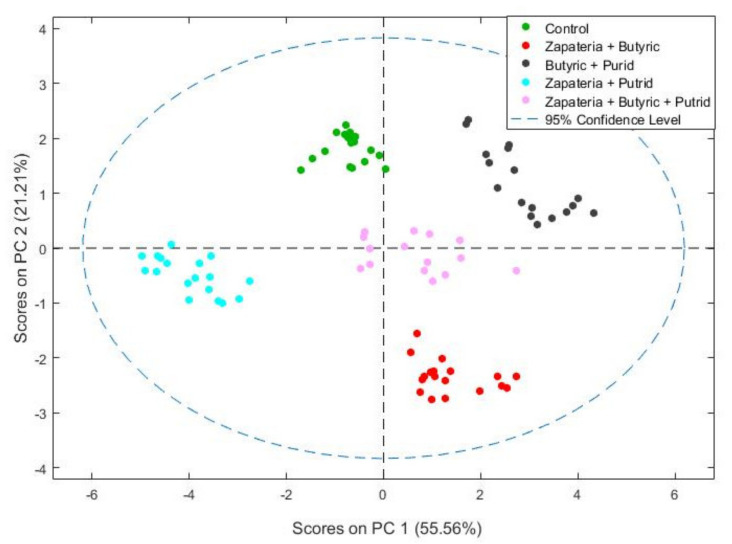
Score plot of the PCA for healthy olives (control) and combinations of zapateria, butyric, and putrid off-odors.

**Table 1 molecules-26-05353-t001:** Predominantly perceived sensory defects of Spanish-style table olive and of combined samples.

	Control	D1	D2	D3	D4
Sensory Evaluation	n.d.	Zapateria	Butyric	Putrid	Musty
		6.0 ± 0.9	5.5 ± 0.7	5.8 ± 0.8	6.0 ± 0.9
	**Control**	**D1 + D2**	**D2 + D3**	**D1 + D3**	**D1 + D2 + D3**
Sensory Evaluation	n.d.	Zapateria	Putrid	Putrid	Putrid
		3.5 ± 0.8	3.5 ± 0.9	3.5 ± 0.8	3.0 ± 0.7

n.d., not detected.

**Table 2 molecules-26-05353-t002:** Relative contents of volatile compounds (mean % (*n* = 3)) obtained from table olives with zapateria, butyric, putrid, and musty defect compared to healthy olives (control). RT, retention time.

	RT (min)		Content (% of Total Area of Identified Compounds)				
		Attributes	Control	Zapateria	Butyric	Putrid	Musty
Carboxylic Acids							
Acetic acid	2.7	Pungent, sour	7.4 ± 0.7	2.9 ± 0.9	2.5 ± 0.7	n.d.	n.d.
Propanoic acid	4.9	Rancid, cheesy	n.d.	4.0 ± 0.5	4.9 ± 0.4	15.8 ± 2.5	0.5 ± 0.1
Butanoic acid	8.2	cheesy	n.d.	37.9 ± 4.4	55.5 ± 4.4	n.d.	1.1 ± 0.1
Pentanoic acid	13.5	Pungent, rancid	n.d.	5.4 ± 1.1	18.5 ± 1.7	n.d.	n.d.
Hexanoic acid	18.5	Pungent, rancid	n.d.	8.5 ± 0.9	n.d.	n.d.	n.d.
(E)-3-Hexenoic acid	20.7	Cheesy, green, dairy	n.d.	17.8 ± 2.3	n.d.	n.d.	n.d.
Cyclohexanecarboxylic acid	26.5	Fruity, woody	n.d.	3.7 ± 0.6	n.d.	n.d.	n.d.
Benzoic acid	28.2	Pungent, sour	7.4 ± 0.7	2.9 ± 0.9	2.5 ± 0.7	n.d.	n.d.
**Alcohols**							
Isopropyl alcohol	1.8	Pungent solvent	n.d.	n.d.	n.d.	16.0 ± 2.8	n.d.
Butan-2-ol	2.4	Winey	n.d.	n.d.	7.6 ± 0.9	n.d.	n.d.
Butan-1-ol	3.1	Fusel, oily	n.d.	n.d.	n.d.	n.d.	2.6 ± 1.0
3-methyl-butan-1-ol	4.7	Woody, whiskey, sweet	n.d.	n.d.	n.d.	9.3 ± 2.1	1.1 ± 0.1
2-methyl-butan-1-ol	4.8	Winey, spicy	n.d.	n.d.	n.d.	5.3 ± 1.6	0.8 ± 0.1
(Z)-3-Hexen-1-ol	9.7	Green, leaf, nuts	0.6 ± 0.1	n.d.	n.d.	n.d.	n.d.
Cyclohexanol	11.3	Camphoreous	n.d.	n.d.	n.d.	0.7 ± 0.1	n.d.
Benzyl alcohol	19.5	Floral, fruity	2.3 ± 0.1	n.d.	n.d.	0.8 ± 0.1	1.0 ± 0.1
Octan-1-ol	21.1	Waxy, green	0.8 ± 0.1	n.d.	n.d.	n.d.	4.6 ± 1
Phenylethyl Alcohol	23.3	Mild rose	7.8 ± 1.0	n.d.	0.9 ± 0.1	15.9 ± 0.6	1.9 ± 0.2
**Phenols**							
Phenol	17.7	Phenolic, plastic	n.d.	n.d.	n.d.	1.7 ± 0.3	0.6 ± 0.1
2-methoxy-phenol	21.9	Smoky, woody, phenolic	3.2 ± 0.3	1.3 ± 0.1	n.d.	3.3 ± 0.4	46.2 ± 6.5
4-ethyl-phenol	26.7	Wet horse, Phenolic	8.7 ± 1.3	n.d.	n.d.	n.d.	n.d.
Creosol	27.0	Spicy	40.2 ± 4.1	4.9 ± 0.2	3.6 ± 0.1	5.8 ± 0.2	0.7 ± 0.1
2,6-Bis(1,1-dimethylethyl)-4-(1-oxopropyl) phenol	38.0	n.d.	n.d.	3.6 ± 0.4	n.d.	n.d.	n.d.
**Aldehydes**							
Octanal	17.3	Fatty, sharp	0.50 ± 0.02	n.d.	n.d.	0.30 ± 0.01	0.40 ± 0.01
**Esters**							
Propyl propionate	7.3	Fruity, berry	0.60 ± 0.01	n.d.	n.d.	n.d.	n.d.
Methyl pentanoate	7.9	Fruity, sweet	n.d.	1.5 ± 0.1	1.4 ± 0.1	n.d.	n.d.
3-Methylbutyl acetate	10.6	Fruity, sweet	0.40 ± 0.01	n.d.	n.d.	1.3 ± 0.1	n.d.
Ethyl pentanoate	11.8	Fruity, fresh	n.d.	1.0 ± 0.1	0.1 ± 0.1	n.d.	n.d.
Methyl hexanoate	13.1	Fruity, pineapple	n.d.	1.3 ± 0.1	1.1 ± 0.1	n.d.	n.d.
3-Methylbutyl propanoate	15.5	Fruity, apricot	n.d.	n.d.	n.d.	0.50 ± 0.01	n.d.
Propyl pentanoate	17.0	Ethereal, fruity	n.d.	0.7 ± 0.1	0.40 ± 0.01	n.d.	n.d.
Ethyl hexanoate	17.1	Sweet, fruity	n.d.	1.0 ± 0.1	0.10 ± 0.01	n.d.	n.d.
4-Hexen-1-ol, acetate	17.4	Fruity	1.2 ± 0.1	n.d.	n.d.	n.d.	n.d.
Propyl hexanoate	22.1	Berry, fruit	n.d.	n.d.	0.20 ± 0.01	n.d.	n.d.
Methyl benzoate	22.2	Herb, lettuce	n.d.	2.3 ± 0.3	n.d.	n.d.	n.d.
Ethyl cyclohexanecarboxilate	24.0	Aromatic, fruity	n.d.	n.d.	n.d.	0.8 ± 0.1	n.d.
n-Propyl benzoate	30.7	Fruity	n.d.	n.d.	n.d.	0.9 ± 0.2	n.d.
**Ketones**							
6-methyl-5-hepten-2-one	16.3	Fruity, pungent, green	0.60 ± 0.01	n.d.	n.d.	n.d.	n.d.
2-Buten-1-one, 1-(2,6,6-trimethyl-1,3-cyclohexadien-1-yl) (damascenone)	35.5	Floral	0.40 ± 0.01	n.d.	n.d.	n.d.	n.d.
**Other Compounds**							
2,4-dimethyl-heptane	6.7	Unpleasant odor of plastic	1.1 ± 0.1	n.d.	n.d.	14.5 ± 2.6	23.6 ± 3.6
Styrene	11.2	Floral, sweet	0.6 ± 0.1	n.d.	n.d.	n.d.	8.7 ± 1.5
1-chlorooctane	20.3	n.d.	n.d.	n.d.	n.d.	n.d.	2.2 ± 0.1
3-ethyl-4-methyl-pyridine	21.6	Sharp, penetrating, strong aromatic odor	n.d.	n.d.	n.d.	2.1 ± 0.4	n.d.
2-Ethenyl-1,1-dimethyl-3-methylene-cyclohexane	23.0	n.d.	15.3 ± 1.0	n.d.	n.d.	n.d.	n.d.
1,2-dimethoxybenzene	24.8	Vanilla	n.d.	n.d.	n.d.	n.d.	2.0 ± 0.2
3,4-dimethoxytoluene	29.2	n.d.	n.d.	n.d.	0.3 ± 0.1	2.1 ± 0.2	n.d.
Copaene	35.2	Woody, spicy	1.1 ± 0.2	n.d.	0.5 ± 0.1	0.8 ± 0.1	0.4 ± 0.1
α-Muurolene	40.4	Woody	n.d.	n.d.	0.10 ± 0.01	0.10 ± 0.01	n.d.
α-Farnesene	40.7	Soft cooking of vegetables, woody	0.1 ± 0.1	n.d.	n.d.	1.7 ± 0.4	1.5 ± 0.2
sum			100.3	100.7	100.2	99.7	99.9

n.d., less than 0.1%.

**Table 3 molecules-26-05353-t003:** Confusion matrix obtained through PLS-DA for discrimination between control (healthy olives) and isolated defects. Values are expressed in percentage.

Predicted Class					
Real Class	Control	Zapateria	Butyric	Putrid	Musty
Control	20	0	0	0	0
Zapateria	0	20	0	0	0
Butyric	0	0	20	0	0
Putrid	0	0	0	19	0
Musty	0	0	0	1	20

**Table 4 molecules-26-05353-t004:** Confusion matrix obtained through PLS-DA for discrimination between control (healthy olives) and combined defects. Values are expressed in percentage.

Predicted Class					
Real Class	Control	Zapateria + Butyric	Butyric + Putrid	Zapateria + Putrid	Zapateria + Butyric + Putrid
Control	20	0	0	0	0
Zapateria + Butyric	0	20	0	0	0
Butyric + Putrid	0	0	20	1	0
Zapateria + Putrid	0	0	0	19	0
Zapateria + Butyric + Putrid	0	0	0	0	20

## Data Availability

All relevant data are included within the manuscript. The raw data are available on request from the authors.

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
