# Peer review of "E-Nose Discrimination of Abnormal Fermentations in Spanish-Style Green Olives"

_molecules, 2021, doi:10.3390/molecules26175353_

Round 1

Reviewer 1 Report

The study is interesting and the data is promising. According to the title "E-nose discrimination of abnormal fermentations in Spanish-style green olives", what is the procedure for practical use of E-nose to discriminate abnormal fermentations in olives? The author should develop a discriminant model or a regression model to judge the quality of the Spanish-style green olives using the E-nose data in present study.

Author Response

Thanks for your appreciation. We agree with the reviewer. We have only included the exploratory analysis through PCA, however the classification model was not included in the manuscript. Thus, taking into account the comment of the referee and to improve the results of the paper, a supervised classification model by using the PLS-DA algorithm was built and included in the new version of the manuscript. Two new tables (Tables 4 and 5) were included in sections 2.4 and 2.5 which contain the confusion matrices for the two PLS-DA models.

Reviewer 2 Report

Comments to the Author:

The authors of this manuscript present an interesting research study regarding the use of novel technologies like the use of E-nose discrimination of abnormal fermentations in Spanish-style green olives. Introduction and material and methods are well described. Also, results are presented in tables and are clear and quite explicable. Due to limited research I believe that the authors discuss and explain the findings of their work. The text needs very few revisions. Although research studies in quality on fruits and vegetables are well described in the past the use of E-nose discrimination of abnormal fermentations in Spanish-style green olives can add further interest.

Abstract

COMMENT:

The abstract is clear and

Introduction

Line 35          I suggest writing:  Olive products produced in Spain…..instead of Spain

Very clear, and not too long, the Hypothesis and purpose of this work is clear too.

Materials and Methods

Lines 299-304     I think that it would be more useful if you describe first the apparatus. In other words, I suggest to start the section with this paragraph instead of paragraph in line 290.

Results and Discussion

Lines 91-99    In my opinion this paragraph could be added to the introduction section. However, if you think that is more suitable in the materials and methods section you could leave it here.

Line 139     Title of the table is in another page

Conclusions

Very clear

Acknowledgments:

Line 340       M Dolores ??

References

COMMENT: Please check the reference list according to the author’s instructions.

Do not capitalize first letters in references 1,16

Check Latin names Saccharomyces cerevisiae in reference 22

In spite of the manuscript is very clear and carefully written some improvement it should be done.

Author Response

Comments to the Author:

The authors of this manuscript present an interesting research study regarding the use of novel technologies like the use of E-nose discrimination of abnormal fermentations in Spanish-style green olives. Introduction and material and methods are well described. Also, results are presented in tables and are clear and quite explicable. Due to limited research, I believe that the authors discuss and explain the findings of their work. The text needs very few revisions. Although research studies in quality on fruits and vegetables are well described in the past the use of E-nose discrimination of abnormal fermentations in Spanish-style green olives can add further interest.

Thanks for your appreciation. We have reviewed the manuscript point by point according to the reviewer.

Abstract

-The abstract is clear

Thanks for your appreciation.

Introduction

-Line 35. I suggest writing:  Olive products produced in Spain…..instead of Spain

Thank you for your appreciation, we changed the sentence.

-Very clear, and not too long, the Hypothesis and purpose of this work is clear too.

Thank you for your comments.

Materials and Methods

-Lines 299-304. I think that it would be more useful if you describe first the apparatus. In other words, I suggest to start the section with this paragraph instead of paragraph in line 290.

Thank you for your observations. The paragraph has been changed according to the proposed upgrades. The changes can be seen in the tracked version.

Results and Discussion

-Lines 91-99. In my opinion this paragraph could be added to the introduction section. However, if you think that is more suitable in the materials and methods section you could leave it here.

We take into account your appreciation and your better understanding, thus we have moved this paragraph to the introduction part. Thank you.

-Line 139. Title of the table is in another page.

We have included together the title of the table with the corresponding text.

Conclusions

Very clear

Thanks.

Acknowledgments:

Line 340. M Dolores ??

We have modified the abbreviation of the name in the acknowledgments epigraph.

References

-Please check the reference list according to the author’s instructions.

We have checked the references according to the author´s instructions.

-Do not capitalize first letters in references 1,16

We have changed this question. Thanks.

-Check Latin names Saccharomyces cerevisiae in reference 22

We have changed this question. Thanks.

-In spite of the manuscript is very clear and carefully written some improvement it should be done.

We have checked the full references in the manuscript.

Round 2

Reviewer 1 Report

The authors have undertaken a serious and successful effort to revise their manuscript according to the additional issues raised by all reviewers. There is no more question.

Author Response

We appreciate the positive comment of Reviewer 1.

He/she did not report any amendment.